# Evidence of a coupled electron-phonon liquid in NbGe$_2$

Hung-Yu Yang [1✉], Xiaohan Yao[1], Vincent Plisson[1], Shirin Mozaffari[2], Jan P. Scheifers [3],
Aikaterini Flessa Savvidou[4], Eun Sang Choi[2], Gregory T. McCandless[3], Mathieu F. Padlewski[5],
Carsten Putzke [5], Philip J. W. Moll [5], Julia Y. Chan [3], Luis Balicas [2,4], Kenneth S. Burch [1] &
Fazel Tafti [1✉]

Whereas electron-phonon scattering relaxes the electron's momentum in metals, a perpetual exchange of momentum between phonons and electrons may conserve total momentum and lead to a coupled electron-phonon liquid. Such a phase of matter could be a platform for observing electron hydrodynamics. Here we present evidence of an electron-phonon liquid in the transition metal ditetrelide, NbGe$_2$, from three different experiments. First, quantum oscillations reveal an enhanced quasiparticle mass, which is unexpected in NbGe$_2$ with weak electron-electron correlations, hence pointing at electron-phonon interactions. Second, resistivity measurements exhibit a discrepancy between the experimental data and standard Fermi liquid calculations. Third, Raman scattering shows anomalous temperature dependences of the phonon linewidths that fit an empirical model based on phonon-electron coupling. We discuss structural factors, such as chiral symmetry, short metallic bonds, and a low-symmetry coordination environment as potential design principles for materials with coupled electron-phonon liquid.

[1] Department of Physics, Boston College, Chestnut Hill, MA, USA. [2] National High Magnetic Field Laboratory, Florida State University, Tallahassee, FL, USA. [3] Department of Chemistry and Biochemistry, University of Texas at Dallas, Richardson, TX, USA. [4] Department of Physics, Florida State University, Tallahassee, FL, USA. [5] Laboratory of Quantum Materials (QMAT), Institute of Materials (IMX), École Polytechnique Fédérale de Lausanne (EPFL), CH-1015, Lausanne, Switzerland. ✉email: yanghw@bc.edu; fazel.tafti@bc.edu

The transport properties of metals with weak electron–electron (el–el) correlations are well described by the Fermi liquid theory and Boltzmann transport equation[1]. Within the standard Fermi liquid theory, the quasiparticles' effective masses, Fermi velocities, and electron–phonon (el–ph) scattering rates can be computed reliably from the first principles. These quantities are then used to calculate the electrical, optical, and thermal properties of metals and semimetals with trivial and topological band structures[2–4] by using the Boltzmann transport equation and assuming momentum-relaxing collisions between electrons and phonons. Historically, a deviation from this standard framework has been predicted if the momentum-relaxing umklapp processes are suppressed so that the momentum transferred from electrons to phonons through el-ph scattering would recirculate from phonons back to electrons (through so-called ph-el scattering) and the total momentum is conserved[5–10].

Recent theoretical works have suggested the emergence of an electron–phonon liquid when not only momentum-relaxing scattering processes such as umklapp and phonon decay are suppressed, but also momentum-conserving scattering is anomalously enhanced through strong ph–el interactions[2,11–14]. The electrical and thermal conductivities of a correlated electron-phonon liquid are predicted to be higher than conventional Fermi liquids due to the momentum-conserving ph–el interactions[11]. In addition, several distinct transport regimes with unconventional thermodynamic properties and hydrodynamic flow are predicted in electron-phonon liquids, but experimental progress is hindered by the lack of candidate materials[12]. Here, we present mounting evidence of such a liquid in the 3D system $NbGe_2$ from three distinct measurements, namely torque magnetometry, electrical and thermal transport, and Raman scattering. Although a subset of these evidence can be found in quasi-2D systems $PdCoO_2$[15,16], $PtSn_4$[17], $WP_2$[2,18], and $WTe_2$[19], such comprehensive evidence of an electron-phonon liquid in a 3D structure has been hitherto missing from the literature. We also discuss the structure-property relationships that lead to the observed behavior and propose design principles to create future 3D candidate materials.

## Quantum oscillations

In a Fermi liquid with weak el–el interactions, density functional theory (DFT) can be used to accurately compute the Fermi surface and effective mass of quasiparticles from the first principles[20]. $NbGe_2$ seems to be just such a system: it is non-magnetic, does not have $f$-electrons, and is not close to a metal-insulator transition. Therefore, it came as a surprise to find out the experimental values of the quasiparticle effective masses ($m^*$) were enhanced consistently beyond the DFT values across all branches of the Fermi surface.

We obtained the experimental $m^*$ values by measuring de Haas-van Alphen (dHvA) effect between 0.5 and 10 K, and from 0 to 41 T. The field was oriented at 41.4° with respect to the hexagonal plane because most of the frequencies were detectable at that angle (Supplementary Fig. 1). The dHvA oscillations and their Fourier transform are plotted in Fig. 1a and b, respectively. The frequency ($F$) of each peak in Fig. 1b is related to the extremal area ($A$) of a closed cyclotron orbit on the Fermi surface through the Onsager relation $F = \frac{\phi_0}{2\pi^2}A$. For every orbit, the quasiparticle effective mass is evaluated by fitting the temperature dependence of the FFT peak intensity to a Lifshitz-Kosevich formula[21,22] (inset of Fig. 1b, Supplementary Fig. 1, and Supplementary Table 1).

The Fermi surface of $NbGe_2$ in Fig. 1c was calculated using density functional theory (DFT) and the theoretical $m^*$ values were obtained using the SKEAF program[23]. A comparison between the theoretical (DFT) and experimental (dHvA) $m^*$ values is presented in Fig. 1d. Both data sets increase uniformly with increasing frequency; however, the experimental values (orange) are three times larger than the theoretical ones (green) at all frequencies. As mentioned above, the el-el interactions must be weak in $NbGe_2$ since it is a non-magnetic metallic system without $f$-electron, and its Fermi surface comprises equal contributions from Ge-$p/s$ and Nb-$d$ orbitals (Supplementary Fig. 2). Thus, the only viable explanation for such a systematic mass enhancement is a strong ph-el interaction.

To put the mass enhancement in perspective, we compare $NbGe_2$ with pure Nb, where detailed dHvA experiments show a 2-fold mass enhancement[24]. Given that less than half of DOS in $NbGe_2$ comes from Nb $d$-orbitals (see Supplementary Fig. 2), the mass enhancement per $d$-level is a factor of 3 larger in $NbGe_2$ than in Nb. Such enhancement can result from either el-el or el-ph interactions. We argue against the former because our DFT calculations in Supplementary Fig. 2 show highly dispersive bands in $NbGe_2$, inconsistent with electronic correlations that are typically associated with flat bands. In fact, flat bands have been observed in $Nb_3Sn$, also with a 2-fold mass enhancement[25,26]. Again, the absence of flat bands in $NbGe_2$ (Supplementary Fig. 2) is inconsistent with electronic correlations. Such comparisons indicate that el-el correlations cannot be responsible for the mass enhancement in $NbGe_2$, suggesting the el-ph coupling as a plausible mechanism. It is also interesting to compare $NbGe_2$ to $WP_2$ and $PdCoO_2$, where strong ph-el interactions and a potential hydrodynamic transport have been evoked[15,16,18]. The effective dHvA masses are less than 1 $m_e$ and 1.5 $m_e$ in $WP_2$ and $PdCoO_2$, respectively[27,28], considerably smaller than $NbGe_2$.

## Electrical resistivity

The second evidence of a coupled electron–phonon liquid in $NbGe_2$ comes from resistivity measurements in Fig. 2. Recent theoretical work has calculated the resistivity curves of $NbGe_2$ by assuming a Fermi liquid ground state, evaluating momentum-relaxing el–ph lifetimes $\tau_{el-ph}^{MR}(\mathbf{k})$ and electron velocities $v_{\mathbf{k}}$ for all bands, and plugging these values into the Boltzmann equation[13]. The resulting theoretical curves are compared to the experimental curves in Fig. 2a and b for in-plane ($\rho_{xx}$) and out-of-plane ($\rho_{zz}$) current directions, respectively. Although the overall anisotropy between the $\rho_{xx}$ and $\rho_{zz}$ channels is consistent between theory and experiment, the theoretical curve within each channel is 6 times larger than the experimental values. As shown in Supplementary Fig. 3, the 6-fold discrepancy persists to low temperatures. Note that the discrepancy is not due to a shortcoming of theory in computing anisotropic scattering rates, since the same calculations correctly capture both the anisotropy and the magnitude of resistivity for $NbSi_2$ and $TaSi_2$[13]. To ensure the discrepancy is not due to uncertainties in sample geometry, we have also measured a standard mesoscopic device (Supplementary Fig. 4) fabricated by focused ion beam (FIB) with geometric uncertainties less than 5% and reproduced the discrepancy.

The 6-fold discrepancy between theory and experiment can arise from el–el, el-defect, or ph–el interactions; however, the first two are ruled out here. Not only the el–el interactions are unlikely in $NbGe_2$, due to the dominance of Ge $p$-orbitals in the band structure (Supplementary Fig. 2), but also they typically lead to a higher experimental resistivity than the theoretical curves, opposite to the observed behavior in Fig. 2a, b. Electron-defect scattering is irrelevant in $NbGe_2$ with a residual resistivity as small as $\rho_{0,xx} = 55\,\text{n}\Omega\,\text{cm}$ along $a$-axis and $\rho_{0,zz} = 35$ $\text{n}\Omega\,\text{cm}$ along $c$-axis (the residual resistivity ratio $RRR > 1000$). Thus, the only plausible source of this discrepancy is the ph-el interaction, which is theoretically predicted to enhance electrical conductivity

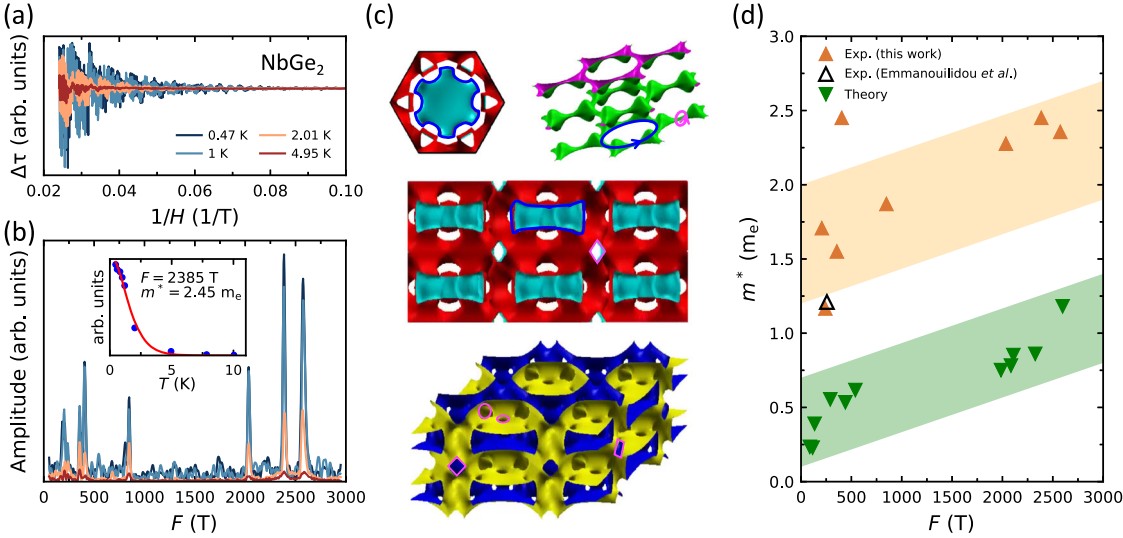

**Fig. 1 de Haas-van Alphen (dHvA) effect. a** de Haas-van Alphen (dHvA) oscillations as a function of inverse field plotted at four representative temperatures. Data were collected from a $NbGe_2$ crystal mounted on a piezoresistive cantilever. **b** The fast Fourier transform (FFT) of dHvA data. The inset shows a Lishitz-Kosevich fit to determine the effective mass at $F = 2358$ T. **c** Calculated Fermi surface of $NbGe_2$. The top left and top right are two representative bands of $NbGe_2$ shown in the first Brillouin zone and repeated zone scheme, respectively. One has a Fermi surface at the center while the other one is hollow. The center shows the side-view of the top left Fermi surface. The bottom Fermi surface is another band similar to the top-left one. The Fermi surface of $NbGe_2$ comprises 4 bands in total, two are like top-left and the other two are like the top-right. The blue and magenta traces are cyclotron orbits as large as 2500 T and as small as 200 T, consistent with observed frequencies. **d** The experimental and theoretical quasiparticle masses plotted as a function of dHvA frequencies showing a three-fold enhancement in the experimental masses. The open data point is reproduced from ref. [37]. The orange and green shaded areas highlight two standard deviations in the experimental and theoretical $m^*$ values, respectively.

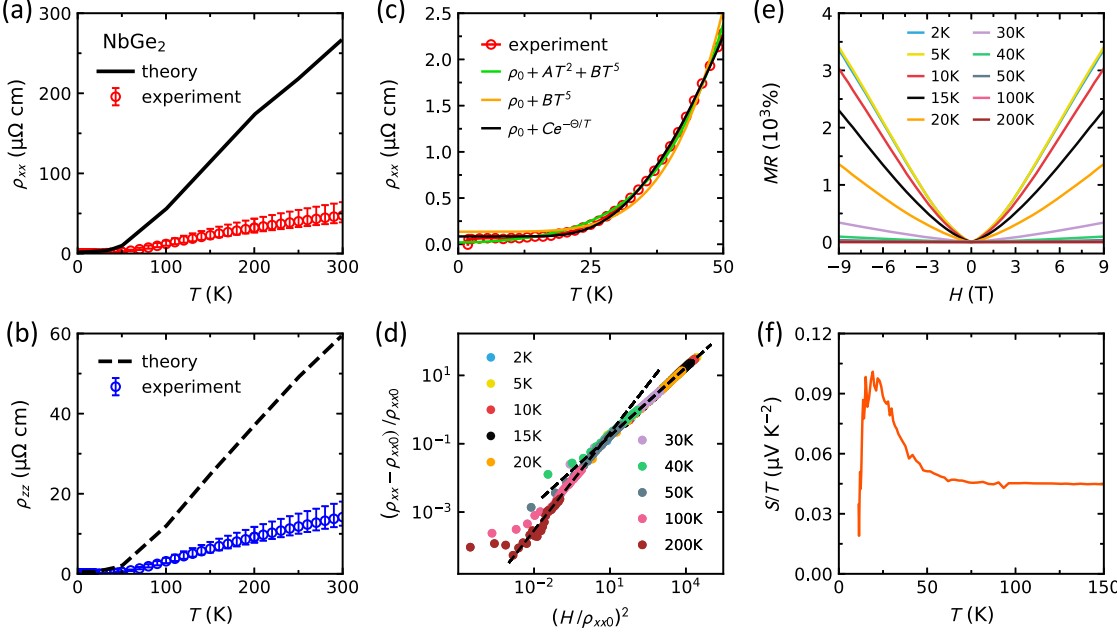

**Fig. 2 Electrical resistivity and thermopower. a** The black lines show theoretical calculations of in-plane resistivity ($\rho_{xx}$) in $NbGe_2$ from ref. [13], and the circles show the experimental data with error bars. The error bars are defined by the finite size of contact pads. The shortest and the longest distance between the pads define the bounds of the error bar. **b** The same comparison is made for the out-of-plane resistivity ($\rho_{zz}$). Both $\rho_{xx}$ and $\rho_{zz}$ are measured on the same $NbGe_2$ sample with $RRR = 1030$. **c** Different models are fitted to the $\rho_{xx}$ data below 50 K. **d** The Kohler scaling analysis shows a change of slope around 50 K. **e** Magnetoresistance MR = $[\rho(H) - \rho_0]/\rho_0$ as a function of the field at several temperatures. **f** Temperature dependence of thermopower ($S/T$) in $NbGe_2$.

beyond a standard Fermi liquid[11], consistent with our observations in Fig. 2a, b. Although the theory has correctly predicted $NbGe_2$ to have an extremely short-lived momentum-conserving phonon-mediated el-el scattering[13], the discrepancy between theory and experiment could indicate an overall underestimation of the momentum-conserving processes in the calculations.

To examine the el–el, el–ph, and ph–el interactions further, we fit three different models to the low-temperature resistivity data ($\rho_{xx}$) in Fig. 2c. Among all models, the black line that represents the phonon-drag model $\rho_{xx} = \rho_0 + Ce^{-\Theta/T}$ yields the best fit (see also Supplementary Fig. 5). This model assumes dominant momentum-relaxing umklapp el–ph scatterings at high-$T$ and

small-angle (quasi momentum-conserving) el–ph and ph–el scatterings at low-$T$[5,28,29]. The fit yields $\Theta = 155$ K, approximately one-third of Debye temperature $\Theta_D = 433$ K determined from the heat capacity measurements in Supplementary Fig. 5. The orange line in Fig. 2c represents the Bloch-Grüneisen model $\rho_{xx} = \rho_0 + BT^5$ that yields a poor fit to the data. Although the fit is improved after adding a $T^2$ el–el scattering term and using $\rho_{xx} = \rho_0 + AT^2 + BT^5$ (green line), the coefficients $A = 2.98 \times 10^{-4}$ μΩ cm K$^{-2}$ and $B = 5.19 \times 10^{-9}$ μΩ cm K$^{-5}$ do not make physical sense. Using the $A$-coefficient of resistivity and the Sommerfeld coefficient from the heat capacity ($\gamma = 6.2$ mJ mol$^{-1}$K$^{-2}$ in Supplementary Fig. 5), we evaluate the Kadowaki-Woods ratio $R_{KW} = \frac{A}{\gamma^2} = 7.7$ μΩ cm mol$^2$K$^2$J$^{-2}$ which is unreasonably large and comparable to the values in heavy fermions (about 10 μΩ cm mol$^2$K$^2$J$^{-2}$)[30]. This is inconsistent with the mild mass renormalization of factor 3 in Fig. 1d and the absence of $f$-electrons in NbGe$_2$.

Based on the above discussion, the phonon-drag $\rho(T)$ behavior in NbGe$_2$ is consistent with a transition from momentum-relaxing umklapp scattering to a momentum-conserving ph–el scattering regime below approximately 50 K. Such a change of scattering length scale is confirmed by a Kohler scaling analysis on the field-dependence of resistivity in Fig. 2d. For this analysis, we use $\rho_{xx}(H)$ curves at 10 different temperatures and plot $[\rho(H) - \rho_0]/\rho_0$ versus $(H/\rho_0)^2$. The curves collapse on a single scaling function that shows a change of slope at approximately 50 K (see the dashed lines in Fig. 2d), consistent with a change of scattering length scale and emergence of an el–ph liquid. The magnetoresistance data (MR = $100 \times (\rho(H) - \rho_0)/\rho_0$) used for the Kohler analysis are shown in Fig. 2e. We have also measured the Seebeck effect (Fig. 2f) and observed an increase of $S/T$ below 50 K followed by a peak at approximately 20 K, consistent with the phonon-drag scenario[11].

## Raman scattering

So far, we have focused on evidence of a correlated electron-phonon liquid in NbGe$_2$ by resorting to the electronic degrees of freedom (transport and dHvA data). Now, we turn to the phononic degrees of freedom by examining the Raman linewidth as a function of temperature in Fig. 3. Temperature-dependent Raman scattering has recently been established as a sensitive tool for revealing the presence of dominant ph–el scattering[18]. In NbGe$_2$, there are 16 modes with the mechanical representation $\Gamma_{opt.} = A_1 + 2A_2 + 3B_1 + 2B_2 + 4E_1 + 4E_2$ that can be detected by Raman. Typically, the finite phonon lifetime (hence finite linewidth) results from the anharmonic decay of optical to acoustic modes. Because phonons are bosons, their linewidths

are expected to scale with the Bose function $n_B(\omega, T)$ and increase with temperature—a behavior well captured by the Klemens model[31,32]. In stark contrast with the Klemens model, however, Fig. 3 shows a non-monotonic temperature dependence in three representative modes that fit a phenomenological model based on phonons decaying into electron-hole pairs. Specifically, the linewidth is given by Fermi (instead of Bose) functions, according to

$$\Gamma(T) \propto n_F(\omega_a, T) - n_F(\omega_a + \omega_0, T) \quad (1)$$

where $\omega_0$ is the phonon frequency, and $\omega_a$ is the energy difference between the electron's initial state and the Fermi energy in a phonon-mediated inter-band scattering[18]. Note that Eq. (1) is entirely phenomenological and independent of a specific theory. Simply put, the $T$-dependences of optical phonons in NbGe$_2$ obey a Fermi (instead of Bose) function, which is possible only if the ph–el scattering dominates ph–ph scattering.

A physical picture of ph–el scattering emerges by comparing the fit parameters $\omega_0$ and $\omega_a$ in the insets of Fig. 3a, b, c. For example, the temperature dependence of the $A_1$ mode in Fig. 3a fits Eq. (1) with $\omega_a \approx \omega_0$, corresponding to a scenario where the initial electronic state is empty at $T = 0$; it begins to populate with increasing temperature, and engages in ph–el scattering into an empty state (hole) via inter-band scattering. In other words, a phonon of frequency $\omega_0$ decays into an el-hole pair. The initial increase of the phonon linewidth is due to increasing ph–el scattering rate with temperature. At higher temperatures, however, the final state (hole) is also populated, so the phonons can no longer decay into an electron-hole pair, and the linewidth decreases. Thus, the initial increase and subsequent decrease of the linewidth is well-captured by Eq. (1) in the entire temperature range. Similar but less pronounced behavior is observed in Fig. 3b for the $E_2(1)$ mode, which fits Eq. (1) but with $\omega_a < \omega_0$. Finally, the behavior in Fig. 3c for the $E_2(2)$ mode is described by Eq. (1) with $\omega_a = 0$, which means the electronic states are already populated at $T = 0$ and the phonon linewidth only decreases with increasing temperature.

## Discussion

A few features in the structural chemistry of NbGe$_2$ may be responsible for the enhanced ph–el coupling in this material. (i) NbGe$_2$ belongs to the C40 structural group, which is chiral due to the presence of a screw axis and the absence of an inversion center (Fig. 4a). Two different chiralities (handedness) are observed among C40 structures[33]; the right-handed CrSi$_2$-type in space group $P6_222$ (#180), and the left-handed NbSi$_2$-type in space group $P6_422$ (#181). The two structures can be

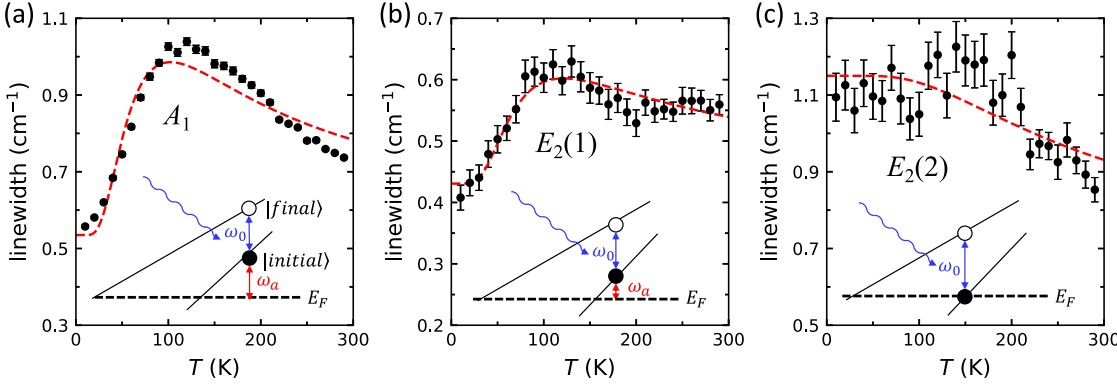

**Fig. 3 Raman scattering.** Temperature dependence of Raman linewidth (proportional to inverse phonon lifetime) is plotted for three optical modes: (**a**) $A_1$, (**b**) $E_2(1)$, and (**c**) $E_2(2)$. The red dashed line is a fit to Eq. (1). For each mode, the relationship between $\omega_0$ and $\omega_a$ (fit parameters in Eq. (1)) is illustrated schematically. Errorbars are set by the full width at half maximum.

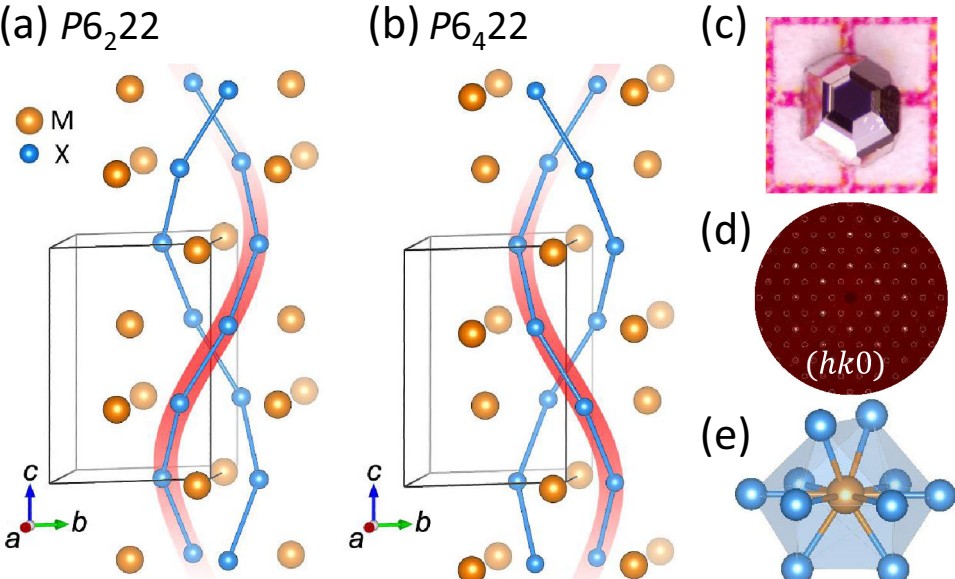

**Fig. 4 Crystal structure of NbGe$_2$. a** The right-handed 6$_2$ and (**b**) left-handed 6$_4$ screw axes are compared in a generic MX$_2$ compound (e.g., NbGe$_2$) by showing the X–X bonds. **c** Picture of a millimeter size NbGe$_2$ crystal. **d** A precession image constructed from the single-crystal X-ray diffraction data. The space group of NbGe$_2$ is $P6_2 22$ (right-handed). **e** The staggered dodecahedral coordination with 10 Ge atoms around each Nb atom in NbGe$_2$.

distinguished by careful single-crystal diffraction experiments (Fig. 4d and Supplementary Table 2). Our crystallographic analysis in the Supplementary Note 5 confirms the right-handed space group $P6_2 22$ in NbGe$_2$ crystals (Fig. 4a). Specifically, a Flack parameter of 0 within the margin of error rules out enantiomeric twinning, which would correspond to the intergrowth of both chiralities (Supplementary Table 2)[34]. Such a well-defined chirality is theoretically proven to stabilize Kramers-Weyl nodes in the electronic band structure[35,36], as confirmed in Supplementary Fig. 2 and elsewhere[13,37]. The Kramers-Weyl nodes may not be relevant to the electronic properties of NbGe$_2$ due to the large Fermi surface and carrier concentration of the order $10^{22}$ el/cm$^3$ (Supplementary Fig. 5 and Supplementary Table 3). However, the lattice chirality may translate into chiral phonon modes which are known to affect transport properties in cuprate materials and control the el-ph coupling in WSe$_2$[38,39]. (ii) The short Nb–Ge and Ge–Ge bond lengths (2.7–2.9 Å) in NbGe$_2$ maximize orbital overlaps and lead to extremely large residual resistivity ratios $RRR > 1000$ and small residual resistivities $\rho_0 < 60$ nΩ cm (Fig. 2 and Supplementary Fig. 7). The large $RRR$ and small $\rho_0$ ensure that the transport signatures of ph-el interactions are not masked by defect scattering. The residual resistivity, carrier concentration ($10^{22}$ el/cm$^3$), and metallic bond lengths in NbGe$_2$ are comparable to those of PdCoO$_2$, which is also a hexagonal system with short Pd-Pd bond lengths of 2.8 Å[40]. PdCoO$_2$ is a candidate of electron hydrodynamics[15,16,41] possibly due to ph-el interactions[14,15,28]. Understanding whether NbGe$_2$ is close to an electron-phonon hydrodynamic regime[12,13] will be an exciting future research direction. (iii) The low-symmetry staggered dodecahedral coordination with 10 Ge around each Nb atom (Fig. 4e) creates nearly isotropic force constants, which in turn promote degenerate phonon states and a bunching between acoustic phonons. It is shown theoretically that such an "acoustic bunching effect" limits the phase space for anharmonic decay of optical to acoustic phonons, leading to the dominance of ph–el over ph–ph scattering[42,43]. A similar effect is likely to suppress anharmonic phonon decays and produce high conductivity as observed in the Weyl semimetal WP$_2$, which also has a low-symmetry coordination environment[2,18]. We propose the combination of a chiral lattice structure, short metallic bonds, and

low-symmetry coordination complex as design principles to create new candidate materials for el–ph liquid[11,12].

Finally, we discuss the significance of an el–ph liquid and contrast it with the old paradigm of phonon drag. In the old literatures[5,6,9], phonon drag is attributed to a loss of momentum-relaxing umklapp el–ph scattering. However, in addition to the suppression of umklapp el–ph scattering, an el–ph liquid must also support (i) strong ph–el interactions that enhance momentum-conserving scatterings, and (ii) strong suppression of momentum-relaxing ph–ph processes. We provide evidence of (i) via quantum oscillations and evidence of (ii) by Raman scattering experiments. The phonon-drag behavior of resistivity and enhanced thermopower support that momentum-conserving scatterings are stronger than momentum-relaxing ones in NbGe$_2$. Points (i) and (ii) go beyond the old paradigm of phonon drag and are specific to an electron-phonon liquid; these two points can potentially conspire with the suppression of umklapp el–ph process to push towards the hydrodynamic limit[2,11–13]. We note that more experiments (such as size-dependent transport experiments) will helpfully establish a coupled el–ph liquid in NbGe$_2$.

## Methods

**Material growth**. Crystals of NbGe$_2$ were grown using a chemical vapor transport (CVT) technique with iodine as the transport agent (see Supplementary Fig. 8). The starting elements were mixed in stoichiometric ratios and sealed in silica tubes under vacuum with a small amount of iodine. We found the best conditions to make high-quality samples was to place the hot end of the tube at 900 °C under a temperature gradient of less than 10 °C, and grow the crystals over a period of one month. Polycrystalline samples were synthesized by heating a stoichiometric mixture of Nb and Ge powders at 900 °C for three days.

**Transport and heat capacity measurements**. The electrical resistivity was measured with a standard four-probe technique using a Quantum Design Physical Property Measurement System (PPMS) Dynacool. The heat capacity was measured using the PPMS with a relaxation time method on a piece of polycrystalline sample cut from sintered pellets. Seebeck coefficient was measured using a one-heater three-thermometer method. A step-wise increase in the heat was applied to generate the corresponding step-wise thermal gradients. The measurements were performed in Quantum Design PPMS, using a custom probe with external electronics, which allowed in-situ calibration of the thermometers in the presence of exchange gas prior to the thermal measurements under a high vacuum.

**X-ray diffraction**. Single crystal X-ray diffraction data were obtained at room temperature using a Bruker D8 Quest Kappa single-crystal X-ray diffractometer operating at 50 kV and 1 mA equipped with an I$\mu$S microfocus source (Mo-K$_\alpha$, $\lambda = 0.71073$ Å), a HELIOS optics monochromator and PHOTON II detector. The structure was solved with the intrinsic phasing methods in SHELXT[44]. No additional symmetries were found by the ADDSYM routine and the atomic coordinates were standardized using the STRUCTURE TIDY routine[45] of the PLATON[46] software as implemented in WinGX 2014.1[47].

**Raman scattering**. Raman spectra were collected in a backscattering mode using a 532 nm Nd:YAG laser with incident power 200 $\mu$W focused to a spot size of 2 $\mu$m in a Montana Instruments cryo-station[48]. Polarization dependence for symmetry identification was performed via the rotation of a Fresnel rhomb which acts as a half-waveplate. The fitting of the phonon features to extract linewidths were performed using a Levenburg-Marquardt least-squares fitting algorithm. Phonons were fit using a Voigt profile, wherein a Lorentzian representing the intrinsic phonon response is convoluted with a Gaussian to account for any broadening induced by the system

**de Haas-van Alphen (dHvA) experiment**. The magneto-quantum oscillation experiments under continuous fields up to 41 T were performed at the National High Magnetic Field Laboratory in Tallahassee, Florida. Temperature and angular dependences of the oscillations were examined to reveal the effective mass and dimensionalities of the Fermi surfaces of the samples. The de Haas-van Alphen effect in the magnetic torque was measured using the piezoresistive cantilever technique (Piezo-resistive self-sensing 300 × 100 $\mu$m cantilever probe, SCL-Sensor.Tech.). A $^3$He cryostat in combination with a rotating probe was used for high-field experiments at temperatures down to 0.35 K.

**Density functional theory (DFT) calculations**. DFT calculations using the linearized augmented plane-wave (LAPW) method were implemented in the WIEN2k code[49] with the Perdew-Burke-Ernzerhof (PBE) exchange-correlation potential[50] plus spin-orbit coupling (SOC). The basis-size control parameter was set to RK$_{max} = 8.5$ and 20000 k-points were used to sample the k-space. Using DFT calculations as input, the Supercell K-space Extremal Area Finder (SKEAF) program[23] was applied to find dHvA frequencies and effective masses of different Fermi pockets.

## Data availability

The data generated in this study have been deposited in the Materials Data Facility (MDF)[51] database under accession code [10.18126/uftm-ny12].

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

## Acknowledgements

F.T. thanks D. Broido, A. Levchenko, and A. Lucas for helpful discussions. F.T. and H.-Y.Y. acknowledge funding by the National Science Foundation under Award No. NSF/DMR-1708929. Work done by V.P. was supported by the US Department of Energy (DOE), Office of Science, Office of Basic Energy Sciences under award no. DE-SC0018675. K.S.B. is grateful for the support of the Office of Naval Research under Award number N00014-20-1-2308. J.Y.C. acknowledges funding by the National Science Foundation under Award No. NSF/DMR-1700030. L.B. is supported by the US-DOE, BES program through award DE-SC0002613. The National High Magnetic Field Laboratory is supported by the National Science Foundation through NSF/DMR-1644779 and the State of Florida. C.P. and P.J.W.M. were supported by the European Research Council (ERC) under the European Union's Horizon 2020 research and innovation program (grant agreement No 715730).

## Author contributions

H.-Y.Y. and X.Y. grew the crystals, carried transport, and heat capacity measurements, and performed DFT calculations. S.M., A.F.S., and L.B. performed the dHvA measurements. E.S.C. measured the Seebeck effect. J.P.S., G.T.M., and J.Y.C. analyzed the single-crystal X-ray diffraction and crystal orientation. M.F.P., C.P., and P.J.W.M. fabricated the FIB device. V.P. and K.S.B. performed Raman scattering. H.-Y.Y. and F.T. wrote the manuscript.

## Competing interests

The authors declare no competing interests.
