## [Peer Review File · Nature Communications]

REVIEWER COMMENTS

Reviewer #1 (Remarks to the Author):

The authors investigate the electronic properties of NbGe₂, a transition metal ditellurides with topological band character. Their main claim of the paper is that the material behaves like an electron-phonon liquid. In such a fluid the total momentum of the combined electron-phonon fluid is conserved at sufficiently low temperatures. This drastically changes the low-T resistivity. It would indeed be very interesting to identify genuine non-local hydrodynamic flow behavior of such an electron-phonon fluid.

The authors first give evidence from dHvA and Raman scattering that there is a sizable electron phonon coupling. In addition, deviations in the resistivity from a quantitative Bloch-Grüneisen analysis (Ref.[15]) for the same material are used to come to the same conclusion.

Overall the issue of electron hydrodynamics is a very timely and interesting topic. However, I have two concerns with this manuscript which the authors must address before publication could be considered.

1. Electron-phonon fluid behavior in the resistivity is not really a new concept. It used to be called phonon drag and has been observed in a range of materials (from delafossites, carbon nano tubes, GaAs heterostructures...). Reviews on phonon drag have been written in the late 1980s. So what is really new in their finding that goes qualitatively beyond the significant class of other phonon drag systems? I could see that the topological band structure of the material gives rise to novel phenomena that have not been explored. To my surprise no substantive discussion of this is made in the manuscript. Another option would be to use FIB-shaped samples discussed in the supplementary material and demonstrate non-local hydrodynamic flow or anomalous resistance noise.

2. I am not fully convinced that the authors actually identified an electron-phonon fluid:

a. Crucial evidence for electron-phonon fluid behavior is usually the low-T activated resistivity dependence $\rho = \rho_0 + A \exp(-\Theta/T)$. Could the authors plot their data from Fig.2c in the regime below approx. 30K in an Arrhenius plot? I simply cannot tell from the current figure whether the black curve is indeed the best fit and whether one can exclude the other, powerlaw dependencies within the error bars.

b. The authors argue that the discrepancy between their measurements and the theory of Ref.15 is evidence for electron-phonon fluid behavior. This discrepancy is most evident for $T > 100\text{K}$ (see Fig. 2a,b). However the authors obtain $\Theta = 155\text{K}$ for the activation scale of the drag resistivity. Above Θ the cited argument of Ref.[8] does not work anymore (the authors implicitly say so themselves). Hence, this reasoning cannot account for the discrepancy between theory and experiment. An alternative reasoning could be the rather anisotropic electron-phonon scattering on the Fermi surface determined in Ref.[15]. This would short circuit the transport (i.e. yield smaller resistivities), while the relaxation time approximation used in Ref.[15] would miss the effect. If this is the correct interpretation, it would provide no support for electron-phonon fluid behavior.

c. The strongest argument in favor of phonon drag behavior is usually the anomalous T-dependence of the thermopower. It would be very interesting if the authors could present such measurements. This may be an unfair request and it should be understood as a friendly suggestion rather a requirement.

d. Why don't the authors observe a size dependence of the resistivity in mesoscopic FIB samples presented in the SM? This would be quite natural in systems that contain hydrodynamic electron fluids. Have they compared the sample size and the scattering length?

Reviewer #2 (Remarks to the Author):

In this paper, Yang et al. present magnetic torque, resistivity and Raman spectroscopy measurements to support their claim that NbGe₂ is a coupled electron-phonon liquid. The work is original and interesting, and the paper is written clearly and concisely. I believe it will be of interest to the readers of Nature Communications. I therefore recommend this article for publication, and have only a few suggestions to the authors.

It is certainly remarkable that the observed effective masses are considerably larger than the theoretically predicted ones. Did the authors only measure magnetic torque oscillations with the magnetic field along one crystallographic direction, and if so, what is it? I believe they should add this information to their figures. In the Emmanouilidou et al. paper that the authors cite, those researchers also observed Fermi pockets with effective masses as small as $\sim 0.2 m_e$, so on the same

order of magnitude as what is theoretically expected. I believe it is worth briefly commenting on that in the manuscript for completeness.

Reviewer #3 (Remarks to the Author):

In this paper the authors have discussed transport and structural properties of NbGe₂ and argue that a coupled electron-phonon liquid is realized in this material. I agree that exploring the possibility of electron-phonon liquid is indeed an interesting topic. However, it is a pity that I cannot see any smoking gun throughout the paper.

1, Before arguing the three fold enhancement of m^* is an evidence of electron-phonon liquid, the authors should discuss to what extent this enhancement is stronger than that of other materials with similar structure. Because ele-ph enhancement to effective mass is a general aspect of all crystalline materials.

2, Smaller resistivity than that of calculation is also discussed as an evidence of electron phonon liquid. But in most cases ab initio calculation of resistivity is difficult because the relaxation events in real materials are hard to capture.

3, Without scrutinizing the electronic structure, excluding correlation effect as an origin of the larger m^* is a bit arbitrary. For example, correlated electrons are recently discussed for twisted graphene.

4, A slope change at 50 K in resistivity is always seen in various material, it cannot be a good evidence too.

5. Because ele-phonon coupling is a general feature of all crystalline materials, in my opinion in order to argue an enhanced ele-phon coupling, one should choose some kinds of standard for comparing, otherwise it is hard to argue 'enhancement'.

5/11/2021

We thank the reviewers for their constructive comments. In response to the reviewers, we have clarified the comparisons between theory and experiments, added Seebeck data to confirm phonon drag, and compared NbGe_2 to other benchmark materials to put our findings in perspective. Our point-by-point response to reviewers' comments and a summary of changes to the manuscript are provided below. All authors agree that the manuscript has significantly improved by including the referees' expert suggestions.

Reviewer #1

"The authors investigate the electronic properties of NbGe_2 , a transition metal ditellurides with topological band character. Their main claim of the paper is that the material behaves like an electron-phonon liquid. In such a fluid the total momentum of the combined electron-phonon fluid is conserved at sufficiently low temperatures. This drastically changes the low-T resistivity. It would indeed be very interesting to identify genuine non-local hydrodynamic flow behavior of such an electron-phonon fluid.

The authors first give evidence from dHvA and Raman scattering that there is a sizable electron phonon coupling. In addition, deviations in the resistivity from a quantitative Bloch-Grüneisen analysis (Ref.[15]) for the same material are used to come to the same conclusion. Overall the issue of electron hydrodynamics is a very timely and interesting topic. However, I have two concerns with this manuscript which the authors must address before publication could be considered."

We appreciate the interest of reviewer #1 in this topic and address their concerns below.

1. "Electron-phonon fluid behavior in the resistivity is not really a new concept. It used to be called phonon drag and has been observed in a range of materials (from delafossites, carbon nano tubes, GaAs heterostructures...). Reviews on phonon drag have been written in the late 1980s. So what is really new in their finding that goes qualitatively beyond the significant class of other phonon drag systems? I could see that the topological band structure of the material gives rise to novel phenomena that have not been explored. To my surprise no substantive discussion of this is made in the manuscript. Another option would be to use FIB-shaped samples discussed in the supplementary material and demonstrate non-local hydrodynamic flow or anomalous resistance noise."

The new aspects of this work are: (1) we make the first comprehensive case for an el-ph liquid based on transport data, mass enhancement of dHvA oscillations, and temperature dependence of Raman line shapes.

(2) we present a *tunable* family of materials, where the electron-phonon coupling can be tuned by selecting $M=\text{Nb,Ta}$ and $X=\text{Si,Ge}$ in MX_2 , (3) transition metal ditetralides are 3D metals unlike carbon nanotubes (1D), GaAs heterostructure (2D), and delafossites (quasi-2D), and (4) our discussion around the structure and coordination chemistry of NbGe_2 lays out a general guideline to find more materials in this regime. We have modified the introduction section to emphasize these points.

The topological aspects (Kramers-Weyl points) are surely interesting but are not stressed deliberately here, because the Weyl nodes are buried in a large Fermi surface in NbGe_2 and we do not expect them to play a role in electron-phonon liquids. The hydrodynamic flow in the quasi-2D Weyl semimetal WP_2 is also mainly discussed in the context of electron-phonon coupling and do not seem relevant to its topological characters [see PRB **98**, 115130 (2018)]. A proper study of nonlocal transport effects requires fabrication of several FIB devices with different geometries and careful cross checks. Such efforts are ongoing, but they are time consuming and deserve a dedicated project.

2. “I am not fully convinced that the authors actually identified an electron-phonon fluid:”

a. “Crucial evidence for electron-phonon fluid behavior is usually the low-T activated resistivity dependence $\rho = \rho_0 + A \exp(-\Theta/T)$. Could the authors plot their data from Fig.2c in the regime below approx. 30K in an Arrhenius plot? I simply cannot tell from the current figure whether the black curve is indeed the best fit and whether one can exclude the other, powerlaw dependencies within the error bars.”

To confirm the low-T activated behavior, we have plotted the data at low temperatures in a semi-log scale as requested by the referee. In Fig. A, the black curve lies within the error bars of nearly all data points, while the other two fits do not. This figure is now added to the supplementary information as Fig. S5a .

Figure A: Resistivity of NbGe_2 as a function of temperature in a semi-log scale.

b. “The authors argue that the discrepancy between their measurements and the theory of Ref.15 is evidence for electron-phonon fluid behavior. This discrepancy is most evident for $T > 100\text{K}$ (see Fig. 2a,b). However the authors obtain $\Theta = 155\text{K}$ for the activation scale of the drag resistivity. Above Θ the cited argument of Ref.[8] does not work anymore (the authors implicitly say so themselves). Hence, this reasoning cannot account for the discrepancy between theory and experiment. An alternative reasoning could be the rather anisotropic electron-phonon scattering on the Fermi surface determined in Ref.[15]. This would short circuit the transport (i.e. yield smaller resistivities), while the relaxation time approximation used in Ref.[15] would miss the effect. If this is the correct interpretation, it would provide no support for electron-phonon fluid behavior.”

Figure B: Comparisons between the calculations of Ref. [24] and our data are presented for (a) ρ_{xx} and (b) ρ_{zz} .

Following the referee's thoughtful comment, we compared our data and the calculations of Ref. [15] (now Ref. [28]) at low temperatures. As shown in Fig. B, the 6-fold discrepancy between experiment and theory is not limited to high temperatures and continues to low temperatures. The original figure was indeed not clear, and we are adding Fig. B in the supplementary information to address this issue.

We argue that the anisotropic electron-phonon scattering does not serve as an alternative reasoning here. The authors of Ref. [28] have indeed computed *anisotropic* el-ph relaxation rates and found that ρ_{xx} is 4.4 times larger than ρ_{zz} in NbGe₂. This result agrees with the experimental data. However, the experimental values of both ρ_{xx} and ρ_{zz} are 6 times larger than the theoretical calculations. Thus, the anisotropic el-ph scattering accounts for the difference *between* ρ_{xx} and ρ_{zz} channels but it does not account for the 6-fold discrepancy that appears *within* each channel. Note that the real material could have disorder, so it is hard to understand how the theory *over* estimates the resistivity. We have clarified this point in the revised manuscript under Section 3.

c. "The strongest argument in favor of phonon drag behavior is usually the anomalous T-dependence of the thermopower. It would be very interesting if the authors could present such measurements. This may be an unfair request and it should be understood as a friendly suggestion rather a requirement."

We followed the referee's advice and measured the thermopower of NbGe₂ as a function of temperature. As seen in Fig. C, S/T starts to increase below 50 K and reveals a peak at approximately 20 K, which is consistent with the phonon-drag behavior at low temperature. These data are now added to Fig. 2f in the manuscript.

Figure C: Seebeck coefficient (S/T) as a function of temperature shows a phono-drag peak at 20 K.

d. “Why don't the authors observe a size dependence of the resistivity in mesoscopic FIB samples presented in the SM? This would be quite natural in systems that contain hydrodynamic electron fluids. Have they compared the sample size and the scattering length?”

The present manuscript is focused on presenting transition metal ditetralides as promising candidates for the electron-phonon liquid. Proving electron hydrodynamics is a major undertaking beyond the scope of this work. A series of devices must be fabricated and many cross checks will be necessary to identify a hydrodynamic regime. Such efforts are under way, but they are time consuming and deserve a separate study.

Reviewer #2

“In this paper, Yang et al. present magnetic torque, resistivity and Raman spectroscopy measurements to support their claim that NbGe₂ is a coupled electron-phonon liquid. The work is original and interesting, and the paper is written clearly and concisely. I believe it will be of interest to the readers of Nature Communications. I therefore recommend this article for publication, and have only a few suggestions to the authors.”

We appreciate the referee's positive assessment of our work.

“It is certainly remarkable that the observed effective masses are considerably larger than the theoretically predicted ones. Did the authors only measure magnetic torque oscillations with the magnetic field along one crystallographic direction, and if so, what is it? I believe they should add this information to their figures. In the Emmanouilidou et al. paper that the authors cite, those researchers also observed Fermi pockets with effective masses as small as $\sim 0.2 m_e$, so on the same order of magnitude as what is theoretically expected. I believe it is worth briefly commenting on that in the manuscript for completeness.”

The detailed temperature dependence of quantum oscillations to evaluate the effective masses was measured with the field at 41.4° with respect to the hexagonal plane, because most of the frequencies were best visible at that angle. The α and α' frequencies with effective masses of $0.2 m_e$ in Ref. [22] were observed with the field in-plane along [100] direction, hence the difference. In Ref. [22], as the field direction changes from in-plane toward c -axis, α and α' are replaced by a new frequency β . We also observe the β frequency (Fig. 1d) consistent with Ref. [22]. Following the referee's advice, we are adding the angle information in the revised manuscript (Section 2) and supplementary information (Fig. S1e).

Reviewer #3

“In this paper the authors have discussed transport and structural properties of NbGe₂ and argue that a coupled electron-phonon liquid is realized in this material. I agree that exploring the possibility of electron-phonon liquid is indeed an interesting topic. However, it is a pity that I cannot see any smoking gun throughout the paper.”

The significance of this work is to present all potential signatures of an el-ph liquid in NbGe₂, including enhanced quasiparticle masses, enhanced conductivity compared to theoretical expectations, phonon-drag peak in thermopower, change of slope in the Kohler scaling analysis, and non-monotonic temperature dependence of Raman linewidth in all optical modes. Although a limited subset of these evidence has been previously reported in WP₂, PdCoO₂, and graphene, there is no other material to show all the evidence collectively.

1. “Before arguing the three fold enhancement of m^* is an evidence of electron-phonon liquid, the authors should discuss to what extent this enhancement is stronger than that of other materials with similar structure. Because ele-ph enhancement to effective mass is a general aspect of all crystalline materials.”

The best comparison would be between NbGe₂ and elemental Nb, for which detailed dHvA data are available [J. Low T. Phys. **30**, 389 (1978)]. The enhancement of the effective mass over the band mass in Nb metal is at most a factor of 2. This enhancement has been attributed to the interaction between d -orbitals (d - d interactions). In NbGe₂, however, less than half of the density of states at E_F are from Nb d -orbitals, and the majority of states come from Ge p -orbitals where electronic correlations are negligible (see Fig. S2). However, the mass enhancement in NbGe₂ over the theoretical values is a factor of 3. Given that only 50% of DOS in NbGe₂ comes from the Nb d -orbitals, the mass enhancement per d -level in NbGe₂ is at least 3 times larger than what would have been expected from el-el correlations. Such a mass enhancement from ph-el interactions is indeed unprecedented.

None of the other candidate materials for strong ph-el interactions (e.g. WP₂ and PdCoO₂) have effective masses larger than $1.5 m_e$ [PRB **96**, 121107 (2017)] and [PRL **109**, 116401 (2012)]. Specifically, PdCoO₂ also has a hexagonal structure similar to NbGe₂, but the mass renormalization is small in that material. Following the referee’s comment, we are adding a discussion of these comparisons to the manuscript under Section 2 and a new panel to Fig. S2.

2. “Smaller resistivity than that of calculation is also discussed as an evidence of electron phonon liquid. But in most cases ab initio calculation of resistivity is difficult because the relaxation events in real materials are hard to capture.”

The first-principles calculations by Garcia *et al.* (Ref. [28] in the revised manuscript) compute the full anisotropic tensor of the el-ph relaxation rates before calculating the resistivity curves. Using this approach, the same authors have previously demonstrated a perfect agreement between theoretical and experimental resistivity curves in WP₂, another topological semimetal with considerable el-ph coupling [PRB **98**, 115130 (2018)]. Among the MX₂ compounds (M=Nb/Ta, X=Ge/Si), their calculations have captured the resistivity of NbSi₂ and TaSi₂ correctly, and our recent experimental data show a decent agreement between their calculations and the resistivity curves of TaGe₂ (Fig. D). Thus, the 6-fold discrepancy in NbGe₂ is truly unusual, and does not reflect inadequacy of the theoretical methods.

Figure D: Comparing the calculations of Ref. [28] to our resistivity data in TaGe₂. The agreement is better than in NbGe₂.

3. “Without scrutinizing the electronic structure, excluding correlation effect as an origin of the larger m^* is a bit arbitrary. For example, correlated electrons are recently discussed for twisted graphene.”

DFT calculations have demonstrated flat bands in twisted graphene due to electronic correlations [PNAS **108**, 12233 (2011)]. There are no such flat bands in NbGe₂, as seen in Fig. S2 of our work and in the band structure calculations of Ref. [28]. Flat bands have been observed in Nb₃Sn, which can be compared to NbGe₂. Detailed dHvA data are available for Nb₃Sn [PRL **40** 1590 (1978)]. Nb₃Sn has a majority of Nb d -levels at E_F with a 2-fold mass enhancement. Whether we use Nb metal or Nb₃Sn as the benchmark, the mass enhancement in NbGe₂ is anomalously large. We added these remarks under Section 2.

4. “A slope change at 50 K in resistivity is always seen in various material, it cannot be a good evidence too.”

This point seems to refer to our Fig. 2d, where a change of slope is observed in the Kohler scaling analysis. To our knowledge, such a behavior is not observed in other metals. For example, WP₂ is a potential candidate for phonon drag and possibly hydrodynamic transport, but the Kohler scaling does not reveal a change of slope (Fig. 3(b) in [PRB **97**, 245101 (2018)]). Note that we are not relying on the change of slope in a single $\rho(T)$ curve, but on a change of slope in the *scaling analysis* of $\rho(H)$ curves at 10 different temperatures. We appreciate the referee pointing this out. The scaling analysis was not explained clearly in the original version. We have clarified this analysis by modifying the last paragraph of section 3.

5. “Because ele-phonon coupling is a general feature of all crystalline materials, in my opinion in order to argue an enhanced ele-phon coupling, one should choose some kinds of standard for comparing, otherwise it is hard to argue 'enhancement'”

Please note that the crucial interaction for an electron-phonon liquid is the ph-el (not el-ph) coupling, i.e. the recycling of momentum from phonons to electrons (not the conventional el-ph scattering). The referee’s point about comparing NbGe₂ to other materials is well taken. In response to points 1, 3, and 4, we have compared

NbGe₂ to elemental Nb, Nb₃Sn, WP₂, PdCoO₂, and graphene. These comparisons are now added under Section 2 in the revised manuscript. We thank the referee for the excellent suggestions and positive criticism.

List of changes to the manuscript

All changes are highlighted in red in the main manuscript and supplementary information.

Title: unchanged

Authors: Eun Sang Choi is added. He has measured the Seebeck effect.

Abstract: minor change in the last sentence

Introduction: The significance of the work is better explained.

Results (quantum oscillations): Information regarding the angle is given. Comparisons are made between NbGe₂ and elemental Nb and Nb₃Sn.

Results (electrical resistivity): The anisotropy between ρ_{xx} and ρ_{zz} is clarified. The discrepancy between theory and experiment is also clarified. The phonon-drag fit to the resistivity is confirmed by providing a semi-log figure in the SI and providing the Seebeck data in Fig. 2e. The Kohler scaling analysis is better explained.

Results (Raman scattering): unchanged

References: 10 new references are added [10, 11, 12, 15, 23, 24, 25, 26, 30, and 50].

Figures: Figure 2e contains new Seebeck data

Methods: The Seebeck measurement is explained

Supplementary Information: Angle dependence of quantum oscillations is shown in Fig. S1e, comparison between theory and experiments is extended to low temperatures in Fig. S3, and the Arrhenius analysis is presented in Fig. S5a.

REVIEWERS' COMMENTS

Reviewer #2 (Remarks to the Author):

The authors have addressed the small issues I raised, and I am happy to recommend this article for publication.

Reviewer #3 (Remarks to the Author):

The authors have substantially revised their manuscript. While a smoking-gun evidence of electron phonon liquid seems still missing, I agree that the discussion presented in this work is of large interest and support publication in Nature Communications.

Remaining comments raised by Reviewer #1:

1) "Electron-phonon fluid behavior in the resistivity is not really a new concept. It used to be called phonon drag... So what is really new in their finding that goes qualitatively beyond the significant class of other phonon drag systems?" This is a VERY critical and important question. Unfortunately, the authors cannot provide clear answer! I still wonder if el-phonon liquid is only a fancy name of the old phenomena.

Indeed, there is a difference between phonon drag and electron-phonon liquid. In the old literatures [Peierls, *Annalen der Physik* 404, 154–168 (1932), Gurevich, *J. Phys. USSR*, 9, 477 (1945), and Gurzhi, *Soviet Physics Uspekhi* 11, 255 (1968)], phonon drag is attributed to a loss of momentum-relaxing Umklapp el-ph scattering. However, in addition to the suppression of Umklapp el-ph scattering, an el-ph liquid must also support (i) strong ph-el interactions that enhance momentum-conserving scatterings, and (ii) strong suppression of momentum-relaxing ph-ph processes. We provide evidence of (i) via quantum oscillations and evidence of (ii) by Raman scattering experiments. The phonon-drag behavior of resistivity and enhanced thermopower support that momentum-conserving scatterings are stronger than momentum-relaxing ones. Points (i) and (ii) go beyond the old paradigm of phonon drag and are specific to an electron-phonon liquid; these two points can potentially conspire with the suppression of Umklapp el-ph process to push towards the hydrodynamic limit [PRB 98, 115130 (2018), *Annals of Physics* 419, 168218 (2020), PRB 103, 155128 (2021), arXiv:2012.09207]. To address this comment, we are adding a few sentences to the introduction and in Sections 3 and 5 to clarify that a coupled el-ph liquid is a new concept beyond the well-established phonon-drag effect.

2) Comment 2a and 2b doubt the fitting to the experimental data. It is hard to judge the significance of these results because there are some unknown pre-factors like "A" and "B". But one point is clear, these fitting and calculations do not form smoking guns. Comment 2c is reasonably addressed by providing the thermoelectric results. Reply to comment 2d also makes sense.

Having added two figures to the supplementary material previously (Figs. S3 and S5), we are afraid that we may not be able to improve points 2a and 2b. In point 2a, the referee wanted to see a magnified view of the comparison between different fits to the $\rho\rho(TT)$ data, which we have provided in Fig. S5. In point 2b, the referee asked whether the difference between theory and experimental curves persist to lower temperatures, which we have confirmed in Fig. S3. Thus, we believe we have adequately addressed those points. We are glad that our responses to comments 2c and 2d have been satisfactory.